# The Electronic Nose Coupled with Chemometric Tools for Discriminating the Quality of Black Tea Samples In Situ

**Shidiq Nur Hidayat** [1], **Kuwat Triyana** [1,2,*], **Inggrit Fauzan** [3], **Trisna Julian** [1],
**Danang Lelono** [3], **Yusril Yusuf** [1], **N. Ngadiman** [4,5], **Ana C.A. Veloso** [6,7] and **António M. Peres** [8,9]

1   Department of Physics, Universitas Gadjah Mada, Yogyakarta 55281, Indonesia
2   Institute of Halal Industry and System (IHIS), Universitas Gadjah Mada, Sekip Utara, Yogyakarta 55281, Indonesia
3   Computer and Electronic Department, Universitas Gadjah Mada, Sekip Utara, Yogyakarta 55281, Indonesia
4   Agricultural Microbiology Department, Universitas Gadjah Mada, Bulaksumur, Yogyakarta 55281, Indonesia
5   PT. Pagilaran, Jl. Faridan M. Noto. 11 Kotabaru, Yogyakarta 55281, Indonesia
6   Instituto Politécnico de Coimbra, ISEC, DEQB, Rua Pedro Nunes, Quinta da Nora, 3030-199 Coimbra, Portugal
7   CEB—Centre of Biological Engineering, University of Minho, Campus de Gualtar, 4710-057 Braga, Portugal
8   Centro de Investigação de Montanha (CIMO), ESA, Instituto Politécnico de Bragança, Campus Sta Apolónia, 5300-253 Bragança, Portugal
9   Laboratory of Separation and Reaction Engineering—Laboratory of Catalysis and Materials (LSRE-LCM), ESA, Instituto Politécnico de Bragança, Campus Santa Apolónia, 5300-253 Bragança, Portugal
*   Correspondence: triyana@ugm.ac.id; Tel.: +62-85-728-15-2871

**Abstract:** An electronic nose (E-nose), comprising eight metal oxide semiconductor (MOS) gas sensors, was used in situ for real-time classification of black tea according to its quality level. Principal component analysis (PCA) coupled with signal preprocessing techniques (i.e., time set value preprocessing, $F1$; area under curve preprocessing, $F2$; and maximum value preprocessing, $F3$), allowed grouping the samples from seven brands according to the quality level. The E-nose performance was further checked using multivariate supervised statistical methods, namely, the linear and quadratic discriminant analysis, support vector machine together with linear or radial kernels (SVM-linear and SVM-radial, respectively). For this purpose, the experimental dataset was split into two subsets, one used for model training and internal validation using a repeated K-fold cross-validation procedure (containing the samples collected during the first three days of tea production); and the other, for external validation purpose (i.e., test dataset, containing the samples collected during the 4th and 5th production days). The results pointed out that the E-nose-SVM-linear model together with the $F3$ signal preprocessing method was the most accurate, allowing 100% of correct predictive classifications (external-validation data subset) of the samples according to their quality levels. So, the E-nose-chemometric approach could be foreseen has a practical and feasible classification tool for assessing the black tea quality level, even when applied in-situ, at the harsh industrial environment, requiring a minimum and simple sample preparation. The proposed approach is a cost-effective and fast, green procedure that could be implemented in the near future by the tea industry.

**Keywords:** electronic nose; black tea; preprocessing; multivariate statistical tools

## 1. Introduction

Tea (*Camellia sinensis*) is the most consumed non-alcoholic beverage in the world, after water [1–10]. Tea can be made from leaf and bud of *Camellia sinensis* through a series of production, fixation, withering, rolling, fermentation, polling and drying processes [4,8,9]. Tea contains a complex chemical composition, depending on its quality from several parameters, like climate, agronomic practices and post-harvest treatment and storage conditions [7]. The phenotype for determining quality levels of black tea in this factory refer to the Indonesian National Standard SNI 1902:2016 that consist of general quality requirements (physical and organoleptic) and specific quality requirements. The organoleptic requirements include color, shape, size and aroma of the tea samples. Meanwhile, the second requirements include the level of polyphenol, ash content and contamination of heavy metals and microbes. For routine and rapid application, the experts usually use only the physical and organoleptic. Consequently, the assessment of tea quality is a hard task that must take into account visual, aroma and taste attributes, which usually are evaluated by trained experts of sensory panels following strict regulations [11]. Nevertheless, quality assessment of tea is a key factor in strengthening confidence between tea producers and consumers.

In tea production, the factory that following a standard of the Indonesian National Standards (SNI 1902:2016). This standard adopts and harmonizes the International Organization for Standardization (ISO 3720)—Black Tea Definition and Basic Requirements [12]. The mandatory control of the implementation in the national markets is performed by Indonesia National Agency of Drug and Food Control (BPOM). The quality level is commonly assessed by trained panelists. However, this analysis, although being the official method, tends to be subjective and depends on the experience of the panelists. Furthermore, the low number of samples that can be evaluated per day poses an additional limitation. Consequently, this procedure, besides being expensive and time-consuming, may lead to inaccurate tea quality classifications [1,2,13,14]. On the other hand, several analytical techniques may be used for tea analysis such as, gas chromatography (GC), capillary electrophoresis (CE), high-performance liquid chromatography (HPLC) and plasma atomic emission spectrometry. These procedures give wide-ranging and accurate information about tea chemical composition, but they exhibit costly instrumental and operation conditions, require skilled technicians to perform the analysis, and are usually time-consuming techniques that may require complex sample pre-treatments [13,15].

Alternatively, infrared spectroscopy, FTIR spectroscopy and colorimetric sensor arrays for have been applied to identify tea variety [13], the quality of Chinese green teas [5] or to assess dry matter content of tea [4]. One of the tea quality test parameters is through testing of volatile organic compounds (VOCs). Brenet et al. used surface plasmon resonance imaging for sensing VOCs that has become a high-selective sensing device [16]. Also, electronic noses (E-nose), that utilize an array of gas sensors to give a fingerprint response to a given odor [17–21], combined with different multivariate statistical tools have been used for tea quality assessment and tea aroma evaluation of green or black teas from different geographical origins [1,2,22–33]. Recently, the possibility of merging different electronic devices (E-nose, E-tongue and/or E-eye), have been studied aiming to improve the overall classification performances of the single devices [3,34–38]. Similarly, E-nose devices, comprising metal oxide semiconductor (MOS) gas sensors have been successfully applied for assessing black tea flavor grade [39] and quality level [1,2,40]. E-Nose have advantages that rapid non-destructive analysis, adequate sensitivity and relatively low cost [41,42] but E-Nose with MOS sensors have disadvantages that sensor drift, susceptible to poisoning, high power consumption, humidity-dependent signal [41]. Therefore, in the above-mentioned works, the E-nose assays were not carried out at the industrial facilities, thus not allowing for a real-time assessment.

In this study, a lab-made E-nose with eight different MOS gas sensors was developed and used in situ in a harsh industrial environment for real-time classification of black tea samples (from seven brands) according to their quality levels. Unsupervised (principal component analysis, PCA) and supervised (linear and quadratic discriminant analysis, LDA and QDA, respectively; and support vector machines, SVM) pattern recognition statistical tools were used together with feature extraction

(preprocessing) methods. Static preprocessing methods were chosen, namely, the time set value (*F1*), area under curve (*F2*), and maximum value (*F3*) methods. To establish the best supervised multivariate classification models, the dataset was split into a training data subset (for learning and internal validation purposes using a repeated K-fold cross-validation (CV) procedure (with 10-folds and 10 repeats)) and a test data subset (for external validation). The objective of this study is to investigate the ability of the E-nose for classifying the quality of tea in the harsh industrial environment with simple feature extraction method and the best classification model performance that could be implemented in the near future of the tea industry.

## 2. Materials and Methods

### 2.1. Tea Sample Preparation

The dried black tea samples, made from fermented leaves or combination of leaves and buds depending on the brand, were obtained from a tea plantation and factory in Pekalongan area (7°11′26.8′′ S, 109°37′35.4′′ E), Central Java, Indonesia. Details regarding the samples, including sample codes, quality level, sample visual aspect, as well as the sample lots (i.e., referring to the production date) included in the training and testing datasets for evaluating the supervised statistical E-nose models established are provided in Table 1.

**Table 1.** List of dried black tea samples.

| Samples Codes | Quality Level | Brand | Sample Visual Aspect | Train Dataset | Testing Dataset |
|---|---|---|---|---|---|
| A1 | Q1 | Broken Orange Pekoe (BOP) |  | D1 D2 D3 | D4 D5 |
| A2 | Q1 | Broken Orange Pekoe Fannings (BOPF) |  | D1 D2 D3 | D4 D5 |
| A3 | Q1 | Pekeo Fanings (PF) |  | D1 D2 D3 | D4 D5 |

**Table 1.** *Cont.*

| Samples Codes | Quality Level | Brand | Sample Visual Aspect | Train Dataset | Testing Dataset |
|---|---|---|---|---|---|
| B1 | Q2 | Pekoe Fanning II (PFF) |  | D1 D2 D3 | D4 D5 |
| B2 | Q2 | Fanning II (FF) |  | D1 D2 D3 | D4 D5 |
| C1 | Q3 | BOHEA |  | D1 D2 D3 | D4 D5 |
| C2 | Q3 | PLUFF |  | D1 D2 D3 | D4 D5 |

As can be seen, tea samples from BOP, BOPF and PF brands belonged to the first (highest) quality level (Q1), while the second (medium) quality level (Q2) included samples from PFF and FF brands. Finally, the third (lowest) quality level (Q3) included the tea samples from BOHEA and PLUFF brands. It should be remarked that the price of the dried tea with a quality level of Q1 is almost twice and four times higher compared to that of Q2 and Q3, respectively.

The tea qualities were determined following the Indonesian National Standards (SNI 1902:2016) and certified by the Rainforest Alliance USA (certificate registration code: RA-F-02638). The first set of samples was obtained in five consecutive days (D1 to D5). For replication purpose, the second to the fifth set of tea samples were collected during the following five days in different weeks. Thus, in total, 175 independent tea samples were collected and further analyzed (7 brands × 5 days × 5 sample independent replicas). For each assay, 2 g of dried tea sample was placed into a 10-mL beaker (sample container) and measured using the E-nose.

## 2.2. Electronic Nose Apparatus and Analysis

A lab-made E-nose with eight MOS gas sensors (Taguchi gas sensor, TGS, series), obtained from Figaro Engineering, Inc. (Osaka, Japan), was used, for the first time, in-situ (at the production line of the tea factory), for real-time analysis of tea samples. The environment temperature and relative

humidity during the measurements were 21 ± 2 °C and 85 ± 10%, respectively. Details regarding each MOS gas sensor comprised in the sensor device as well as the targeted gases that may be detected are given in Table 2. From Table 2, some sensors present cross sensitivity to some chemical compounds that can increase the range of the potential applications of the device in the case of E-nose. Besides the gas sensor array, the built E-nose device comprised a sampling system, a data acquisition unit (DAQ), and a signal-processing framework, as shown in Figure 1. The sampling system comprises two electronic valves (three-way system) for airflow control.

**Table 2.** List of sensors used in the E-nose.

| Main Targeted Analytes | Sensor Series | Measurement Range | Sensitivity (Change Ratio of Rs) | Limit of Detection (LoD), ppm | Cross-Sensitivity |
|---|---|---|---|---|---|
| Alcohol, Solvent Vapors [43] | TGS 2620 | 50 to 5000 ppm ethanol | 0.3 to 0.5 in ethanol | 50 | Methane, carbon monoxide, iso-butane, hydrogen, ethanol |
| Methane, Propane, Iso-Butane [44] | TGS 2612 | 1 to 25% Lower Explosive Limits (LEL) of each gas | 0.50 to 0.65 | 300 | Ethanol, methane, iso-butane, propane |
| Chlorofluorocarbons [45] | TGS 832 | 4 to 40 kΩ in 1,1,1,2-Tetrafluoroethane (R-134a) at 100 ppm/air | 0.50 to 0.65 | 10 | Chlorofluorocarbons, hydrofluorocarbons refrigerant gas, ethanol |
| Organic Solvent Vapors [46] | TGS 822 | 1 to 10 kΩ in ethanol at 300 ppm/air | 0.40 ± 0.10 | 50 | Methane, carbon monoxide, isobutane, n-hexane, benzene, ethanol, acetone |
| Air Contaminants (Trimethylamine, Methyl Mercaptan, etc.) [47] | TGS 2603 | 1 to 10 ppm ethanol | <0.5 | 0.3 | Hydrogen, hydrogen sulfide, methyl mercaptan, trimethylamine, ethanol |
| Air Contaminants (Hydrogen, Ethanol, etc.) [48] | TGS 2600 | 1 to 30 ppm of $H_2$ | 0.3 to 0.6 | 1 | Methane, carbon monoxide, iso-butane, ethanol, hydrogen |
| Combustible Gases [49] | TGS 813 | 5 to 15 kΩ in methane at 1000 ppm/Air | 0.60 ± 0.05 | 500 | Carbon monoxide, methane, ethanol, propane, isobutane, hydrogen |
| Ammonia [50] | TGS 826 | 30 to 300 ppm | 0.55 ± 0.15 | 30 | Iso-butane, hydrogen, ammonia, ethanol |

During the sampling step, airflow (a reference gas) from reference connector to valve 1 pass through the sample container to valve 2 and entering the gas sensor array setup. Meanwhile, during the delay or purging phase, the airflow from the reference gas directly passes through the gas sensor array. A 16-bit analog-to-digital converter (ADC) in an Arduino Mega microcontroller was used for the data acquisition system. Every second, a dataset with 10 signal values is sent from the microcontroller unit to the data logger using RS-232 serial communication.

Prior to measuring, the E-nose was turned for about 30 min to ensure that a steady state response is obtained. Afterward, the configuration of the phase time was set as: 10 s delay phase, 60 s sampling phase, and 300 s for purging phase. The assays were performed in situ, at the factory facilities located in the tea plantation area, without any special conditioning of room were the measurements were carried out, simulating a real application of the E-nose at an industrial level. Before each assay, the headspace equilibrium time was set equal to 5 min. The typical of sensor signal characteristics of the E-nose sensors is illustrated in Figure 2, representing the three stages of the E-nose analysis (the delay phase, the sampling phase, and the purging phase) of the *i*-gas sensor for a *j* sample.

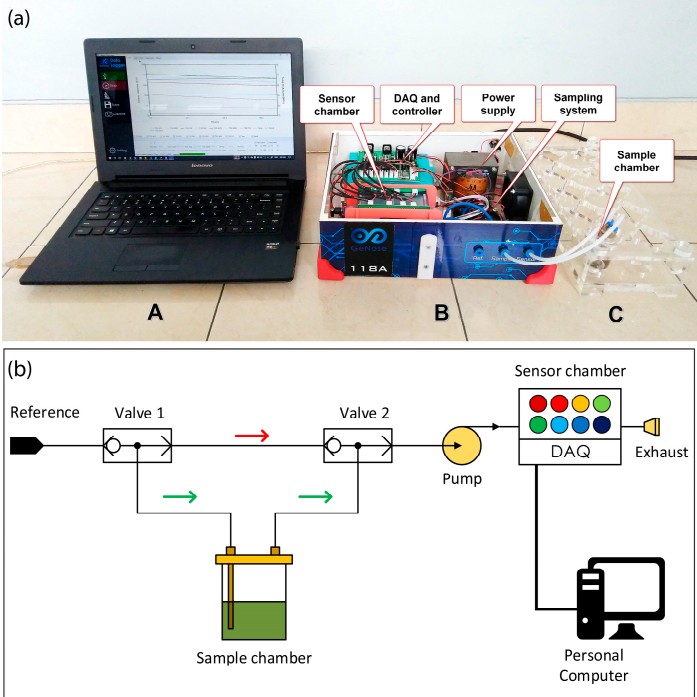

**Figure 1.** (**a**) Photograph the E-nose device used in this study (A: computer with chemometric tools, B: main part of the E-nose devices, and C: chamber of sample); (**b**) Schematic diagram of the E-nose device (DAQ: data acquisition system).

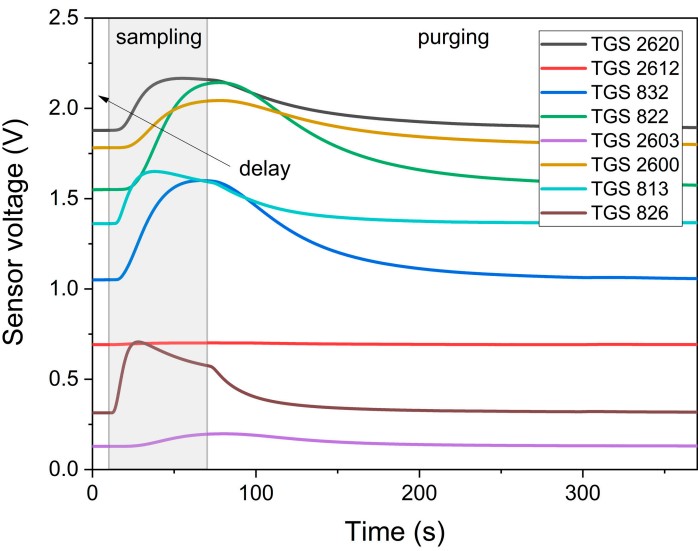

**Figure 2.** An example of the real e-nose responses when detecting a BOP tea sample.

### 2.3. E-nose Raw Signal Profiles and Data Preprocessing

The data matrix used for the statistical multivariate analysis included the E-nose signal profiles gathered by the eight MOS gas sensors during the tea samples analysis. The matrix data of each tea sample contained 8 sensors × 3700-row data. Each sensor $i$ of the E-nose, an electrical signal profile was generated over the analysis time-period, for each $j$ sample ($V_{ij}(t)$). The time-dynamic response of each sensor depends on several physical parameters, such as flow rate in the sampling system, the sample's headspace, the reaction time between the sample volatile compounds and the sensing material, and the environmental conditions like pressure, temperature and humidity [51]. In this work, three types

of data preprocessing techniques were applied namely the time set value (*F*1), the area under the curve (*F*2, as illustrated a shading area in Figure 2, as described in Equation (1)), and the maximum value (*F*3, as described in Equation (2)) preprocessing tools [52]. When required, a time period of 70 s was used, corresponding to the time delay and the sampling analysis time as illustrated in Figure 2.

$$X_{i,j} = \int_{t_0}^{t_1} V_{i,j}(t)dt \tag{1}$$

$$X_{i,j} = \max\left(V_{i,j}(t)\right) \tag{2}$$

### 2.4. Statistical Analysis Using Unsupervised and Supervised Pattern Recognition Tools

The dataset included the preprocessed signals gathered by the eight MOS gas sensors comprised in the E-nose during the 175 assays corresponding to 175 independent samples evaluated (7 brands × 5 days × 5 sample repetitions). Principal component analysis (PCA) was used as an unsupervised pattern recognition method aiming to reduce experimental data dimension [15]. So, PCA would allow a preliminary evaluation of the capability of the E-nose to be used as unsupervised classification model of the tea samples according to their quality level.

The E-nose performance for tea quality assessment, was further evaluated by applying four supervised statistical methods, namely LDA, QDA and SVM with linear and radial kernels (RBF kernel) [53]. Previously to the statistical analysis, the data matrix was subjected to a normalization procedure (scaling and centering techniques). After this normalization step, the initial database was split into two groups, the training data subset and the testing data subset. The training data subset was used to establish the classification model, being selected the one that allowed achieving the best classification performance for the repeated K-fold cross-validation (CV) procedure (10 repeats × 10 folds, which ensured that at each validation run, 10% of the training data was left for internal validation purposes), while the testing data subset was used for the external-validation (full prediction) of the classification model previously established. For SVM, the cost and gamma parameters were tuned to get the best classification performance. The training data subset included the samples collected during the first three days of tea production (i.e., D1, D2, and D3 sets), while the testing data subset included the samples produced during the 4th and 5th production days (i.e., D4 and D5 sets) as shown in Table 1. The overall modeling development and analysis were performed using the open-source statistical software R (version 3.5.1) and using the caret [54], MASS [55], and Kernlab [56] libraries.

## 3. Results and Discussion

### 3.1. Principal Component Analysis: E-nose Performance for Tea Quality Classification

Three PCA models were established, one for each of the E-nose preprocessed data methods used (*F*1, *F*2, and *F*3 data preprocessing techniques). The first five principal components (PCs) functions explained more than 99% of the data variance, accounting the first three PCs of each model for 94.43%, 92.24% and 94.27% of the data variability, for *F*1, *F*2 and *F*3 procedures, respectively. As can be inferred from the 3D plots of the first three PCs (Figures 3–5), the tea samples could be satisfactorily grouped according to the quality level (Q1, Q2 and Q3 grades), showing that the E-nose signals enabled the unsupervised sample classification. The results also pointed out the E-nose signal profiles allowed to closely group tea samples of the same brand (25 samples per brand), although a slight brand overlapping was observed between BOP and BOPF brands with PF brand, FF with PFF brand, and BOHEA with PLUFF brands. This overlapping could indicate that the aroma profiles recorded by the E-nose can be used as a tea quality fingerprint rather than a tea brand fingerprint, meaning that the olfaction device could be a tool for quality assessment tool rather than a brand classifier.

Finally, the reported quality assessment achieved with the E-nose and PCA, is in agreement with the assessment of the tea factory sensory panelists, which classified samples of BOP, BOPF and PF brands as the highest tea quality level (Q1 grade), those of PFF and FF brands as the medium tea quality level (Q2 grade), and finally, the samples from BOHEA and PLUFF brands as the lowest tea quality level (Q3 grade). Thus, the E-nose showed the potential to be used as a preliminary tool for tea quality assessment. Furthermore, from the PCA results it is also possible to infer that the *F*3 data preprocessing method allowed achieving the best-unsupervised discrimination performance.

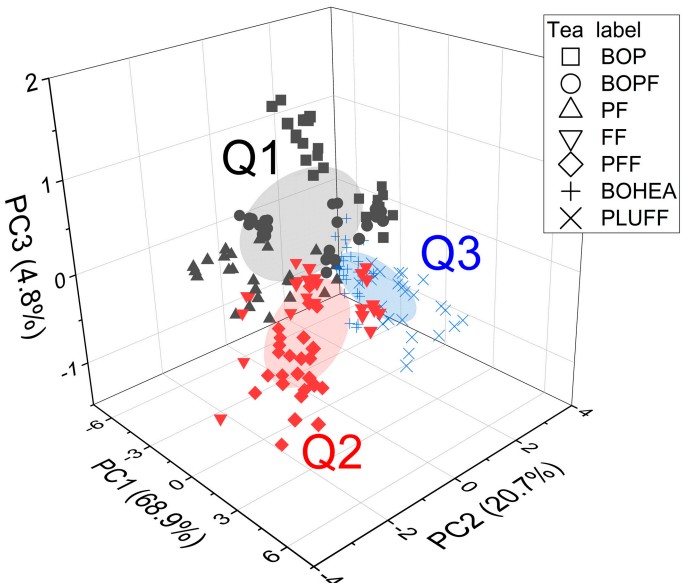

**Figure 3.** PCA 3D plots with confident level 50% regarding the unsupervised classification of tea samples from different brands (BOHEA, BOP, BOPF, FF, PF and PLUFF as listed in Table 1) according to their quality grades (Q1, Q2 and Q3) based on the E-nose profiles after signal preprocessing data treatment using *F*1 procedure.

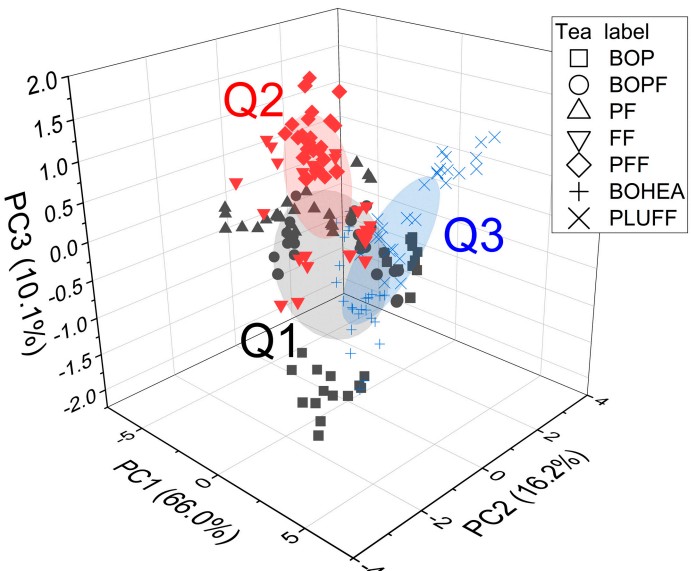

**Figure 4.** PCA 3D plots with confident level 50% regarding the unsupervised classification of tea samples from different brands (BOHEA, BOP, BOPF, FF, PF and PLUFF, as listed in Table 1) according to their quality grades (Q1, Q2 and Q3) based on the E-nose profiles after signal preprocessing data treatment using *F*2 procedure. PC1, PC2 and PC3 are the 1st, 2nd and 3rd principal component functions.

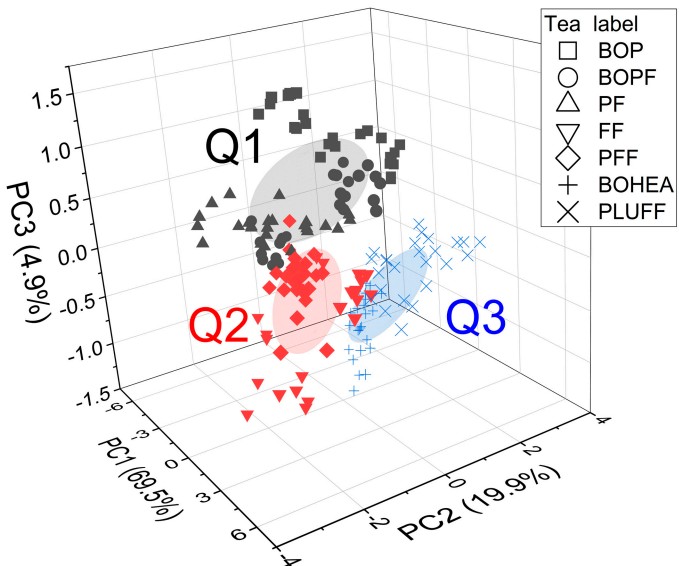

**Figure 5.** PCA 3D plots with confident level 50% regarding the unsupervised classification of tea samples from different brands (BOHEA, BOP, BOPF, FF, PF and PLUFF, as listed in Table 1) according to their quality grades (Q1, Q2 and Q3) based on the E-nose profiles after signal preprocessing data treatment using *F*3 procedure. PC1, PC2 and PC3 are the 1st, 2nd and 3rd principal component functions.

### 3.2. Supervised Multivariate Classification Methods

Supervised multivariate classification methods, including LDA, QDA, SVM linear, and SVM radial, were further used to verify the capability of the E-nose to correctly predict, in a real-time basis and using data collected in-situ, the quality grade of tea samples. First, the multicollinearity between the E-nose preprocessed signals (after applying the *F*1, *F*2 and *F*3 techniques) gathered by the 8 MOS gas sensors was evaluated. The magnitude of the *R*-Pearson coefficients (data not shown) showed that the preprocessed signals of some E-nose sensors (S1, S3, S4, S6 and S7) were collinear ($0.81 \leq R$-Pearson $\leq 0.98$), being the preprocessed signals of sensors S2, S5 and S8 not collinear ($R$-Pearson $\leq 0.69$). The collinearity found is often reported for E-noses [27] and so, different linear and non-linear supervised chemometric tools were used in this study since they can differently deal with collinearity issues [28].

For evaluating the predictive performance of the E-nose coupled with LDA, QDA, SVM linear or SVM radial models, each of the three initial datasets comprising the preprocessing signals (*F*1, *F*2 and *F*3 techniques) were firstly split into two groups. As mentioned, one group (105 tea samples of the seven different brands and three quality levels, collected during the three consecutive production days: D1, D2 and D3), was used for training and internal validation purposes. For the internal validation, the repeated K-fold-CV variant (10 repeats × 10 folds, leading to the 100 classification runs) was used for resampling purposes (ensuring that in each run, 10% of the training data were kept aside for internal validation), allowing estimating, in a more realistic manner (in comparison with the usual leave-one-out CV variant), the potential predictive performance of each model. The second data subset (70 tea samples of the seven different brands and three quality levels, collected during the two other consecutive production days: D4 and D5) was used for testing the predictive capabilities of each model previously established (external validation). It should be remarked that for the SVM-linear model the cost value was set equal to 1, for all the data preprocessing methods. In contrary, for the SVM-radial approach the best values of the two tuning parameters (the cost and the sigma values) were optimized (cost values of 16, 16, and 2, for *F*1, *F*2, and *F*3 methods, respectively; and sigma value equal to 0.2596189, 0.2934098, and 0.2246985, for *F*1, *F*2, and *F*3 methods respectively).

The results (Figure 6) showed that based on the training and internal validation (repeated K-fold-CV) procedures the four supervised pattern recognition tools allowed establishing satisfactory tea quality classification models, based on the signals gathered by the MOS gas sensors of the E-nose

with sensitivities (i.e., percentages of correctly classified samples) greater than 90%. The SVM-linear approach was the one that showed a better agreement between the classification performances for the training and internal validation procedures, pointing out that this technique could be less prone to noise effects. Moreover, the results (Figure 6) also allowed inferring that the *F3* data preprocessing technique allowed achieving the best discrimination rates, independently of the supervised classification technique used. Finally, the full predictive performance of each E-nose-supervised classification model was further evaluated using the external validation data subset, using the best model established for each of the four methods studied (LDA, QDA, SVM-linear and SVM-radial).

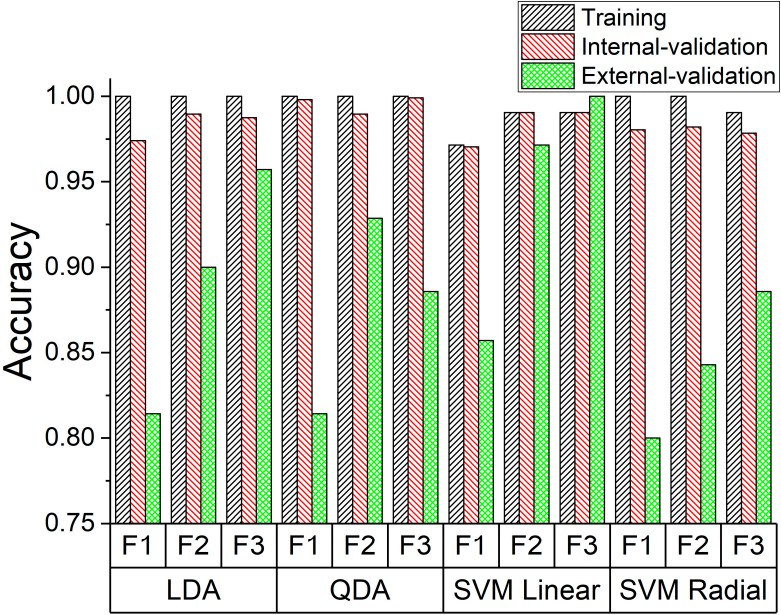

**Figure 6.** Accuracy achieved for training (black bars), internal validation using the repeated K-fold-CV procedure (10 repeats and 10 folds, red bars), and external-validation (testing dataset, green bars) for the combinations of supervised pattern recognition tool (LDA, QDA, SVM-linear or SVM-radial) and data preprocessing method (*F1*, *F2* or *F3*).

As can be inferred from Figure 6, the predictive classification performance of the E-nose-LDA, E-nose-QDA and E-nose-SVM-radial models significantly decreased compared to the performances previously achieved for the internal validation procedure, even if the more robust repeated K-fold-CV variant was applied. In fact, the correct classification rates decreased from values greater than 0.95 to values ranging from 0.80 to 0.93 (external validation). On the contrary, the E-nose-SVM-linear approach enabled obtaining the best predictive performance, with overall correct classification rates of 95% and 100% for *F2* and *F3* data preprocessing methods, respectively. These findings pointed out the relevance of selecting the adequate pattern-recognition multivariate model/data preprocessing technique for achieving the best E-nose classification performance. The sensitivity and specificity values obtained for the external-validation data for each individual tea quality group evaluated, are listed in Tables 3 and 4. The results clearly showed the overall superior performance of the SVM-linear models and the enhanced results achieved by applying the *F3* preprocessing technique. The satisfactory results clearly demonstrated the capability of using, in-situ on a real-time basis, an E-nose with MOS gas sensors, together with the SVM-linear model coupled with the maximum value signal preprocessing method (*F3* technique), as a practical and potential routine analytical tool for correctly assessing tea quality level (Q1, Q2 or Q3 grades), for the seven tea brands studied (BOP, BOPF, PF, FF, PFF, BOHEA, and PLUFF). Furthermore, it should be remarked that, the satisfactory E-nose performance is in line with the discrimination performances previously reported by other researchers (sensitivities ranging from 80 to 100%) using E-nose devices for tea quality classification [1,3,24,26,27,29,32–34,39].

However, it should also be noticed that, contrary to the present study, the majority of these results were obtained for training or internal validation datasets, and only is few cases a test dataset was used to verify the predictive E-nose performances. Moreover, and contrary to this study, in the previous works, the E-nose devices were not tested at the usual harsh industrial conditions located in the tea plantation area.

**Table 3.** Sensitivity values (correct classification rate, i.e., true positive rate) achieved for the external validation dataset (testing dataset), per grouping class.

| Data Preprocessing | Class | LDA | QDA | SVM-Linear | SVM-Radial |
| --- | --- | --- | --- | --- | --- |
| *F1* | Q1 | 0.80 | 1.00 | 1.00 | 0.83 |
| | Q2 | 0.65 | 0.35 | 0.50 | 0.55 |
| | Q3 | 1.00 | 1.00 | 1.00 | 1.00 |
| *F2* | Q1 | 0.90 | 1.00 | 0.97 | 0.90 |
| | Q2 | 0.80 | 0.75 | 0.95 | 0.60 |
| | Q3 | 1.00 | 1.00 | 1.00 | 1.00 |
| *F3* | Q1 | 1.00 | 1.00 | 1.00 | 1.00 |
| | Q2 | 0.85 | 0.60 | 1.00 | 0.60 |
| | Q3 | 1.00 | 1.00 | 1.00 | 1.00 |

**Table 4.** Specificity values (i.e., true negative rate) achieved for the external validation dataset (testing dataset), per grouping class.

| Data Preprocessing | Class | LDA | QDA | SVM-Linear | SVM-Radial |
| --- | --- | --- | --- | --- | --- |
| *F1* | Q1 | 0.85 | 0.68 | 0.75 | 0.78 |
| | Q2 | 0.96 | 1.00 | 1.00 | 0.96 |
| | Q3 | 0.90 | 1.00 | 1.00 | 0.94 |
| *F2* | Q1 | 0.90 | 0.88 | 0.98 | 0.80 |
| | Q2 | 0.98 | 1.00 | 0.98 | 0.98 |
| | Q3 | 0.96 | 1.00 | 1.00 | 0.96 |
| *F3* | Q1 | 0.95 | 0.80 | 1.00 | 0.80 |
| | Q2 | 1.00 | 1.00 | 1.00 | 1.00 |
| | Q3 | 0.98 | 1.00 | 1.00 | 1.00 |

## 4. Conclusions

This study demonstrates the versatility and practical potential of using a lab-made E-nose, with MOS gas sensors, coupled with signal preprocessing methods and chemometric tools for assessing the quality grade of black tea samples, from different brands and different production lots (i.e., production days). The black tea samples were obtained from tea factories in the Pekalongan region of Central Java, Indonesia, being the assays performed in situ, showing for the first time, that an E-nose could be used as a tea quality assessment tool, even in the usual harsh factory environments. The results showed that the maximum value preprocessing method allowed for the best unsupervised and supervised overall correct classification rates. Also, the E-nose-SVM-linear model enabled achieving 100% of correct full predictive classifications (external-validation dataset containing 70 independent tea samples of three quality levels from the seven brands, collected during two production days). Hopefully, the reported satisfactory predictive performance may contribute to the future implementation of this type of E-nose device/chemometric approach by tea factories as well as by governmental agencies, for black tea quality assessment. The E-nose could be used in situ and on a real-time basis as a preliminary fast and cost-effective tea quality assessment tool, reducing the number of samples to be evaluated by expert sensory panelists, overcoming the scarcity of official sensory panels, minimizing the human subjectivity on the tea evaluation, and increasing the limited number of samples panelists can evaluate per day.

**Author Contributions:** Conceptualization, K.T. and A.M.P.; Data curation, I.F., D.L., and N.N.; Formal analysis, S.N.H., K.T., A.C.A.V. and A.M.P.; Funding acquisition, K.T.; Investigation, I.F.; Methodology, K.T., T.J., A.C.A.V.; Project administration, T.J.; Resources, Y.Y. and N.N.; Software, S.N.H., A.C.A.V. and A.M.P.; Supervision, K.T. and N.N.; Validation, K.T., T.J., Y.Y.; Visualization, D.L. and A.M.P.; Writing—original draft, S.N.H., D.L.; Writing—review & editing, K.T., A.C.A.V., A.M.P.

**Funding:** This research and APC were funded by the Ministry of Research, Technology and Higher Education of the Republic of Indonesia through a research scheme of PTUPT 2019 (Contract No. 2688/UN1.DITLIT/DIT-LIT/LT/2019). This work was also financially supported by strategic project UID/EQU/50020/2019—Associate Laboratory LSRE-LCM, strategic project PEst-OE/AGR/UI0690/2014–CIMO, strategic funding UID/BIO/04469/2019-CEB and BioTecNorte operation (NORTE-01-0145-FEDER-000004), all funded by European Regional Development Fund (ERDF) through COMPETE2020—Programa Operacional Competitividade e Internacionalização (POCI)—and by national funds through FCT—Fundação para a Ciência e a Tecnologia I.P.

**Acknowledgments:** The authors would like also to thank PT Pagilaran (tea plantation and factory in Pekalongan area, Central Java, Indonesia) for providing all tea samples.

**Conflicts of Interest:** The authors declare no conflicts of interest.

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
