# Peer review of "The Electronic Nose Coupled with Chemometric Tools for Discriminating the Quality of Black Tea Samples In Situ"

_chemosensors, doi:10.3390/chemosensors7030029_

Reviewer 1 Report

The paper by S.N. Hidayat et al. aims at analyzing the performances of an e-Nose on the discrimination of black tea quality. 

The experiments are done on a relevant amount of tea products on-site which is a huge advantage compared to previous studies.

I would still have some comments:

1/ In introduction it is mentioned the interest of real-time analysis. It should be mentioned that MOS sensors are not the only technique to provide such advantage. For example recently SPR imaging was shown to be relevant for gas detection (Brenet et al., Anal Chem 90, 16, 9879-9887, 2018). This should be also mentioned.

2/ Line 109: a reference should be given if it has been previously described in another paper. 

3/ In Figure 2 it should be added real data to shown the kinetics of the adsorption and desortion of COVs on the sensors.

4/ In the 3D graph of Fig. 3, 4 and 5 it is hard to distinguish the segregation between clusters. It would be useful to add the 2D PCA plots of the two first components.

In conclusion I believe this paper is interesting and deserves publication after taking into account the comments above.

Author Response

Answers to Reviewers’ comments and suggestions

Manuscript ID chemosensors-531184

“Electronic Nose Coupled with Chemometric Tools for in situ Discriminating the Quality Level of Black Tea Samples”

Journal: Chemosensors

We acknowledge the Reviewers for their comments, suggestions and positive remarks which are an incentive for continuing our research work. Following the suggestions of Reviewer #1, Reviewer #2 and Reviewer #3 (comments send in an attached file), changes were made and highlighted with green color (for Reviewer #1), Turquoise (for Reviewer #2) and yellow (for Reviewer #3) in the revised Ms. version.

We would like to thank Prof. Dr. Franz L. Dickert (Handling Associate Editor), Prof. Dr. Giovanni Neri (Handling Associate Editor-in-Chief) and Dr. Igor Medintz (Handling Editor-in-Chief) of Chemosensors for giving us the chance to improve our work.

COMMENTS TO THE AUTHOR:

Reviewer #1: The paper by S.N. Hidayat et al. aims at analyzing the performances of an e-Nose on the discrimination of black tea quality. The experiments are done on a relevant amount of tea products on-site which is a huge advantage compared to previous studies.

Authors answer: We acknowledge the overall positive and kind remarks of the reviewer.

Detailed remarks about the paper are as follows:

1.       In introduction it is mentioned the interest of real-time analysis. It should be mentioned that MOS sensors are not the only technique to provide such advantage. For example recently SPR imaging was shown to be relevant for gas detection (Brenet et al., Anal Chem 90, 16, 9879-9887, 2018). This should be also mentioned.

Authors answer: As suggested by the Reviewer, we added another technique than MOS sensors using the suggestion paper.

The additional sentences can be seen as follows:

In text page 2 line 79–81: “One of the tea quality test parameters is through testing of volatile organic compounds (VOCs). Brenet et.al. used surface plasmon resonance imaging for sensing VOCs that has become a high-selective sensing device [16].”

2.       Line 109: a reference should be given if it has been previously described in another paper.

Author answer: Thank you for pointing that out. We clarified that issue.

The revised sentences can be seen as follows:

In text page 4 line 126-127: “For each assay, 2-g of dried tea sample was placed into a 10-mL beaker glass (sample container) and measured using the E-nose.”

3.       In Figure 2 it should be added real data to show the kinetics of the adsorption and desorption of COVs on the sensors.

Author answer: As suggested by the Reviewer, we changed Figure 2 with the real response of the E-nose to a BOP tea sample.

The changes of Figure 2 and its figure caption can be seen as follows:

Figure 2. An example of the real e-nose responses when detecting a BOP tea sample.

4.       In the 3D graph of Fig. 3, 4 and 5 it is hard to distinguish the segregation between clusters. It would be useful to add the 2D PCA plots of the two first components.

Author answer: As suggested by the Reviewer, we evaluated as follows.

The PCA is an unsupervised pattern recognition method that can reduce the dimensionality of the data by orthogonal transformation into principal components (PCs). It is used mainly as an exploratory data analysis tool since it allows to reveal the internal structure of the data applied to the E-nose signals data matrix in order to evaluate data variability. Visually, the data variability of PCA 3D plot is much higher than that of the 2D plot as shown below.

When using F1 procedure

When using F2 procedure

When using F3 procedure

Linear discriminant analysis (LDA) and other chemometric tools used in this study, on the other hand, are supervised pattern recognition methods to verify the capability of the E-nose to correctly classify the black samples according to the quality. Visually, the 2D plots of LDA are shown below.

When using F1 procedure

When using F2 procedure

When using F3 procedure

From these reasons, therefore, we did not include the 2D plot of PCA and LDA in the manuscript.

Reviewer 2 Report

The proposed method is very interesting and has a large application potential. I believe that after the introduction of minor amendments (see below), this manuscript can be qualified for further stages of evaluation.

line 70-77, Electronic noses have found many uses in the analysis of food, more and more works on this subject appear in various magazines. It would be worth inserting an additional paragraph devoted to the advantage and disadvantages of the electronic nose and to present a wide spectrum of applications of the electronic nose in the analysis of food. I would also like to quote a few works on this subject, eg:

a. Electronic noses in classification and quality control of edible oils: A review. Food Chemistry, 246, (2018), pp. 192-201.

b. A review of gas sensors employed in electronic nose applications, Sensor Review, 24, 181–198, 2004.

c. Electronic nose: current status and future trends, Chemical Reviews, 108, 705–725, 2008.

d. Portable Electronic Nose Based on Electrochemical Sensors for Food Quality Assessment, Sensors 2017, 17(12), 2715.

e. Electronic noses for food quality: A review. J. Food Eng. 2015, 144, 103–111.

f. A compact and low cost electronic nose for aroma detection. Sensors 2013, 13, 5528–5541.

line 78-87. In addition to the application purpose, please also write what is the scientific goal. What does this bring new to science?

line 115-116, I suggest to the sensors given which compounds are sensitive and give the LOD value for these compounds.

figure 3, the first three components explained more than 100% of the data variable (this is an obvious error please check it)

Author Response

Answers to Reviewers’ comments and suggestions

Manuscript ID chemosensors-531184

“Electronic Nose Coupled with Chemometric Tools for in situ Discriminating the Quality Level of Black Tea Samples”

Journal: Chemosensors

We acknowledge the Reviewers for their comments, suggestions and positive remarks which are an incentive for continuing our research work. Following the suggestions of Reviewer #1, Reviewer #2 and Reviewer #3 (comments send in an attached file), changes were made and highlighted with green color (for Reviewer #1), Turquoise (for Reviewer #2) and yellow (for Reviewer #3) in the revised Ms. version.

We would like to thank Prof. Dr. Franz L. Dickert (Handling Associate Editor), Prof. Dr. Giovanni Neri (Handling Associate Editor-in-Chief) and Dr. Igor Medintz (Handling Editor-in-Chief) of Chemosensors for giving us the chance to improve our work.

COMMENTS TO THE AUTHOR:

Reviewer #2: The proposed method is very interesting and has a large application potential. I believe that after the introduction of minor amendments (see below), this manuscript can be qualified for further stages of evaluation.

Authors answer: We acknowledge the overall positive and kind remarks of the reviewer.

Detailed remarks about the paper are as follows:

1.       line 70-77, Electronic noses have found many uses in the analysis of food, more and more works on this subject appear in various magazines. It would be worth inserting an additional paragraph devoted to the advantage and disadvantages of the electronic nose and to present a wide spectrum of applications of the electronic nose in the analysis of food.

Author answer: As suggested by the Reviewer, we added information about it.

The additional sentences can be seen as follows:

In text page 2 line 74, 81-82, 88-91: “Also, electronic noses (E-nose), that utilize an array of gas sensors to give a fingerprint response to a given odor [17],…

E-Nose has advantages that rapid non-destructive analysis, adequate sensitivity and relatively low cost [37,38] but E-Nose with MOS sensors have disadvantages that sensor drift, susceptible to poisoning, high power consumption, humidity-dependent signal [37]. Therefore,…”

2.       line 78-87. In addition to the application purpose, please also write what is the scientific goal. What does this bring new to science?

Author answer: As suggested by the Reviewer, we added information about it.

The additional sentences can be seen as follows:

In text page 3 line 102-105: “The objective of this study is to investigate the ability of the E-nose for classifying the quality of tea in the harsh industrial environment with simple feature extraction method and the best classification model performance that could be implemented in the near future of the tea industry.”

3.       line 115-116, I suggest to the sensors given which compounds are sensitive and give the LOD value for these compounds.

Author answer: As suggested by the Reviewer, we added the information about it by adding Table 2.

The additional of it can be seen as follows:

In text page 4 line 133-137: “Details regarding each MOS gas sensor comprised in the sensor device as well as the targeted gases that may be detected are listed in Table 2. From Table 2, some sensors present cross sensitivity to some chemical compounds that can increase the range of the potential applications of the device in the case of E-nose.”

Table 2. List of sensors used in the E-nose.

Main targeted analytes

Sensor series

Measurement Range

Sensitivity

(change ratio of Rs)

Limit of Detection (LoD), ppm

Cross-Sensitivity

Alcohol, Solvent vapors [43]

TGS 2620

50 to 5000 ppm EtOH

0.3 to 0.5 in ethanol

50

Methane, carbon monoxide, iso-butane, hydrogen,   ethanol

Methane, propane, iso-butane [44]

TGS 2612

1 to 25%LEL of each gas

0.50 to 0.65

300

Ethanol, methane, iso-butane, propane

Chlorofluorocarbons [45]

TGS 832

4kΩ to 40kΩ in R-134a at 100ppm/air

0.50 to 0.65

10

Chlorofluorocarbons, hydrofluorocarbons refrigerant   gas, ethanol

Organic Solvent Vapors [46]

TGS 822

1kΩ to 10kΩ in ethanol at 300ppm/air

0.40 ± 0.10

50

Methane, carbon monoxide, isobutane, n-hexane,   benzene, ethanol, acetone

Air contaminants

(Trimethylamine, methyl mercaptan, etc.) [47]

TGS 2603

1 to 10ppm EtOH

<0.5< p="">

0.3

Hydrogen, hydrogen sulfide, methyl mercaptan,   trimethylamine, ethanol

Air contaminants

(hydrogen, ethanol, etc.) [48]

TGS 2600

1 to 30ppm of H2

0.3 to 0.6

1

Methane, carbon monoxide, iso-butane, ethanol,   hydrogen

Combustible Gases [49]

TGS 813

5kΩ to 15kΩ in methane at 1000ppm/Air

0.60 ± 0.05

500

Carbon monoxide, methane, ethanol, propane,   isobutane, hydrogen

Ammonia [50]

TGS 826

30 to 300 ppm

0.55 ± 0.15

30

Iso-butane, hydrogen, ammonia, ethanol

4.       figure 3, the first three components explained more than 100% of the data variable (this is an obvious error please check it)

Author answer: Thank you for pointing that out. As we carefully investigated the image, we found that we incorrect writing the number of PC3 (48.4%). We corrected to be 4.8%.

The changes in Figure 3 can be seen as follows:

Figure 3. PCA 3D plots regarding the unsupervised classification of tea samples from different brands (BOHEA, BOP, BOPF, FF, PF and PLUFFA, as listed in Table 1) according to their quality grades (Q1, Q2 and Q3) based on the E-nose profiles after signal pre-processing data treatment using F1 procedure.

Reviewer 3 Report

Suggestions for the article entitled "Electronic Nose Coupled with Chemometric Tools for in situ Discriminating the Quality Level of Black Tea Samples"

The proposed research work is well done and of good scientific interest.

I only add a few small suggestions for the authors to optimize this excellent work.

row  50/51: I suggest to talk about phenotype, given the importance of this condition for the real quality of the final product.

row 57, are not just made these types of checks, and respect what legislation? ISO? please list the mandatory and voluntary controls and in which country or continent they are made

row 68/69 where is this technique used on the production line (on line sistems) or in the laboratory?

row 72, have these techniques been probably implemented because they are faster, cheaper and easier to use than classical chemical / physical techniques?

row 98 are the brand names real or are they in acronym?

row 116 I suggest to equip the device with sensors from different manufacturers to improve its capabilities

Fig. 2 I suggest to insert the units of measurement

Author Response

Answers to Reviewers’ comments and suggestions

Manuscript ID chemosensors-531184

“Electronic Nose Coupled with Chemometric Tools for in situ Discriminating the Quality Level of Black Tea Samples”

Journal: Chemosensors

We acknowledge the Reviewers for their comments, suggestions and positive remarks which are an incentive for continuing our research work. Following the suggestions of Reviewer #1, Reviewer #2 and Reviewer #3 (comments send in an attached file), changes were made and highlighted with green color (for Reviewer #1), Turquoise (for Reviewer #2) and yellow (for Reviewer #3) in the revised Ms. version.

We would like to thank Prof. Dr. Franz L. Dickert (Handling Associate Editor), Prof. Dr. Giovanni Neri (Handling Associate Editor-in-Chief) and Dr. Igor Medintz (Handling Editor-in-Chief) of Chemosensors for giving us the chance to improve our work.

COMMENTS TO THE AUTHOR:

Reviewer #3: Suggestions for the article entitled "Electronic Nose Coupled with Chemometric Tools for in situ Discriminating the Quality Level of Black Tea Samples". The proposed research work is well done and of good scientific interest. I only add a few small suggestions for the authors to optimize this excellent work.

Authors answer: We acknowledge the overall positive and kind remarks of the reviewer.

Detailed remarks about the paper are as follows:

1.       row  50/51: I suggest to talk about phenotype, given the importance of this condition for the real quality of the final product.

Authors answer: As suggested by the Reviewer, we added information about phenotype.

The additional sentences can be seen as follows:

In text page 2-3 line 52-58: “The phenotype for determining quality levels of black tea in this factory refers to the Indonesian National Standard SNI 1902:2016 that consist of general quality requirements (physical and organoleptic) and specific quality requirements. The organoleptic requirements include color, shape, size and aroma of the tea samples. Meanwhile, the second requirements such as level of polyphenol and ash content and contamination of heavy metals and microbes. For routine and rapid application, the experts usually use only the physical and organoleptic.”    

2.       row 57, are not just made these types of checks, and respect what legislation? ISO? please list the mandatory and voluntary controls and in which country or continent they are made

Authors answer: As suggested by the Reviewer, we added information about it.

The additional sentences can be seen as follows:

In text page 2 line 63-66: “In the tea production, the factory that following a standard of the Indonesian National Standards (SNI 1902:2016). This standard adopts and harmonizes ISO 3720 - Black tea definition and basic requirements [12]. The mandatory control  of the implementation in the national markets is performed by Indonesia National Agency of Drug and Food Control (BPOM).”

3.       row 78/79 where is this technique used on the production line (on line systems) or in the laboratory?

Authors answer: This technique was used in the same location of the tea production that maybe at the harsh industrial environment.

4.       row 72, have these techniques been probably implemented because they are faster, cheaper and easier to use than classical chemical / physical techniques?

Authors answer: At this moment, this technique has not been implemented in the factory. After the standardization and calibration process, the E-nose will be ready to implement for  routine assessment of the black tea quality in the factory.

5.       row 98 are the brand names real or are they in an acronym?

Authors answer: The brand names are an acronym but for BOHEA and PLUFF are the real names. We added information about the acronym in Table 1.

The additional about it can be seen as follows:

Table 1. List of dried black tea samples

Sample

Quality level

Brand

Sample visual aspect

Train dataset

Testing dataset

A1

Q1

Broken   Orange Pekoe

(BOP)

D1

D2

D3

D4

D5

A2

Q1

Broken   Orange Pekoe Fannings

(BOPF)

D1

D2

D3

D4

D5

A3

Q1

Pekeo   Fanings

(PF)

D1

D2

D3

D4

D5

B1

Q2

Pekoe   Fanning II

(PFF)

D1

D2

D3

D4

D5

B2

Q2

Fanning   II

(FF)

D1

D2

D3

D4

D5

C1

Q3

BOHEA

D1

D2

D3

D4

D5

C2

Q3

PLUFF

D1

D2

D3

D4

D5

6.       row 116 I suggest to equip the device with sensors from different manufacturers to improve its capabilities

Authors answer: Thank you for your suggestion. We will consider it to add other sensors from different manufacturers in the next device.

7.       Fig. 2 I suggest inserting the units of measurement

Authors answer: As suggested by the Reviewer, we changed Figure 2 with the real response of the E-nose to BOP tea samples with the units of measurement (in voltage), that can be seen as follows:

An example of the real e-nose responses when detecting a BOP tea sample.

Round  2

Reviewer 1 Report

The paper is now ready for publication.